# The Relationship between Oral Health-Related Quality of Life and Body Mass Index in an Older Population from Southern Italy: The Salus in Apulia Study

**DOI:** 10.3390/jpm13091300

**Published:** 2023-08-25

**Authors:** Vittorio Dibello, Frank Lobbezoo, Rodolfo Sardone, Madia Lozupone, Fabio Castellana, Roberta Zupo, Alberto Pilotto, Antonio Daniele, Vincenzo Solfrizzi, Daniele Manfredini, Francesco Panza

**Affiliations:** 1Department of Orofacial Pain and Dysfunction, Academic Centre for Dentistry Amsterdam (ACTA), University of Amsterdam and Vrije Universiteit Amsterdam, 1081 HV Amsterdam, The Netherlands; 2Local Healthcare Authority of Taranto, 74121 Taranto, Italy; 3Department of Translational Biomedicine and Neuroscience “DiBraiN”, University of Bari Aldo Moro, 70124 Bari, Italy; 4Department of Interdisciplinary Medicine, Clinica Medica e Geriatria “Cesare Frugoni”, University of Bari Aldo Moro, 70124 Bari, Italy; 5Geriatrics Unit, Department of Geriatric Care, Orthogeriatrics and Rehabilitation, Galliera Hospital, 16128 Genoa, Italy; 6Department of Neuroscience, Catholic University of Sacred Heart, 00168 Rome, Italy; 7Neurology Unit, IRCCS Fondazione Policlinico Universitario A. Gemelli, 00168 Rome, Italy; 8Department of Prosthodontics and Dental Materials, University of Siena, 53100 Siena, Italy

**Keywords:** oral health, body mass index, obesity, OHRQoL, OHIP-14, aging, older people

## Abstract

Background: The assessment of oral health-related quality of life (OHRQoL) evaluated the impact of an individual’s oral health on the patient’s physical and psychosocial status. We evaluated the association between subjective OHRQoL, measured with the Oral Health Impact Profile-14 (OHIP-14) questionnaire, and unfavorable body mass index (BMI) (i.e., too high or too low) in a large population-based study on older adults from Southern Italy. Moreover, we assessed which of the seven OHIP-14 domains was the most strongly associated with an unfavorable BMI. Methods: We used data on a subpopulation of the Salus in Apulia Study, including 216 older adults. BMI < 18.4 kg/m^2^ and >30 kg/m^2^ were classified as unfavorable, while values between 18.5 and 30 kg/m^2^ were classified as ideal. Results: A higher OHIP-14 total score increased the risk of an unfavorable BMI (odds ratio (OR): 1.08, 95% confidence interval (CI): 1.01–1.15). In the model adjusted for age, sex, education, hypertension, carbohydrate consumption, and alcohol consumption, this finding was confirmed with a higher OHIP-14 total score increasing the risk of an unfavorable BMI (OR: 1.10, 95% CI: 1.01–1.22), and higher age linked to a decreased risk of an unfavorable BMI (OR: 0.89, 95% CI: 0.82–0.97). In a random forest regression model, the most important predictive domains/sub-scales of OHIP-14 in the mean decrease in the Gini coefficient for unfavorable BMI were, in order of decreasing importance, physical pain, functional limitation, psychological discomfort, physical disability, social disability, psychological disability, and handicap. Conclusions: In older age, negative OHRQoL, particularly linked to the physical pain domain, increased the risk of being underweight or overweight and obesity.

## 1. Introduction

According to the World Health Organization [1], oral health is a key indicator of overall health, well-being, and quality of life. Several studies have shown that oral and general health are strongly interlinked [2,3,4,5]. In fact, systemic disorders such as cardiovascular diseases [4], cancer [5], chronic respiratory diseases [2], and diabetes mellitus [3] share common modifiable risk factors with most oral diseases and conditions. Tobacco use, alcohol consumption, and diets high in free sugars are among these risk factors, all of which are increasing at the global level [6].

Furthermore, deteriorating oral health, especially in older age, together with a reduction in oral hygiene, may lead to a progression of caries and periodontal disease resulting in tooth loss, which can in turn lead to changes in diet and nutritional health [7]. The reduction in the number of teeth is accompanied by different food choices as partially or fully edentulous patients tend to prefer softer over hard foods, which may have lower nutritional values. Macro- and micronutrient deficiencies resulting from these nutritional imbalances are linked to functional impairment in both underweight and overweight older adults, increasing the risk of falls, fractures, infections, frailty, and dementia [8,9,10,11].

However, an assessment of oral clinical indicators alone is often not adequate to correctly describe health status, especially concerning emotional aspects. On the other hand, it has been reported that individuals with chronic debilitating diseases can consider their quality of life to be higher than healthy people, implying that bad health or sickness does not always reflect a low quality of life [12,13]. In recent years, the assessment of oral health-related quality of life (OHRQoL) has been widely used to evaluate the impact of an individual’s oral health on the patient’s physical and psychosocial status, including a self-assessment of emotional well-being, expectations, and therapeutic satisfaction, becoming a relevant component of chronic disease management [13,14]. One of the most widely used tools for assessing OHRQoL is the Oral Health Impact Profile-14 (OHIP-14) questionnaire, including 14 items within seven domains related to functional limitation, physical pain, psychological discomfort, physical, psychological, social disability, and handicap [15]. This is a shorter version of the OHIP-49 [16]. A recent study examined the nutritional characteristics of older adults and their relationship to OHRQoL, measured by the Geriatric Oral Health Assessment Index (GOHAI), showing that individuals with a poor perception of their oral health were more likely to have an unfavorable body mass index (BMI) [17], i.e., BMI ≤ 18.4 kg/m^2^ (underweight), BMI between 25.0 and 29.9 kg/m^2^ (overweight), and BMI ≥ 30.0 kg/m^2^ (obese). However, at present, there is a lack of evidence on whether a negative OHRQoL may have an impact on BMI changes. The first aim of the present study was to evaluate the association between subjective OHRQoL, measured with the OHIP-14, and unfavorable BMI in a large population-based study on older adults from Southern Italy. The second aim was to assess which of the seven domains of the OHIP-14 questionnaire (namely functional limitation, physical pain, psychological discomfort, physical disability, psychological disability, social disability, and handicap) was the most strongly associated with higher variations in BMI and, therefore, with an increased clinical occurrence of underweight or overweight/obesity in older age.

## 2. Materials and Methods

### 2.1. Study Sample and Design

Data used in this cross-sectional study were from the “Salus in Apulia Study”, a public health initiative funded by the Italian Ministry of Health and Apulia Regional Government, and conducted at the Istituto di Ricovero e Cura a Carattere Scientifico (IRCCS) “S. De Bellis”, National Institute of Gastroenterology and Research Hospital, Castellana Grotte, Bari, Italy. The sampling framework was the regional office list on 31 December 2014, which included 19,675 individuals, 4021 of whom were 65 years or older. The study design and data collection method are described in detail elsewhere [18,19]. The present study used data on a subpopulation of the Salus in Apulia Study, including 216 older adults who agreed to participate by answering survey questions on their OHRQoL. Before their assessment, all subjects signed informed consent, and the study was authorized by the Institutional Review Board of the IRCCS “S. de Bellis”, National Institute of Gastroenterology and Research Hospital, Castellana Grotte, Bari, Italy (approval Code: 68/CE De Bellis; approval Date: 9 April 2019). The study was conducted following the Declaration of Helsinki in 1975 and followed the STARD (Standards for Reporting Diagnostic Accuracy Studies) and STROBE (Strengthening the Reporting of Observational Studies in Epidemiology) guidelines.

### 2.2. Assessment of the Oral Health-Related Quality of Life (OHRQoL)

For the assessment of OHRQoL, the Italian version of the OHIP-14 [16,20] was used, which is a shorter version of the OHIP-49 [15,21], designed to measure self-reported dysfunction, discomfort and disability attributed to oral conditions. The questionnaire is based on a conceptual oral health model outlined by Locker [22]. The original instrument has 49 items, representing 7 domains (functional limitation, physical pain, psychological discomfort, physical disability, psychological disability, social disability, and handicap). The short-form version of the OHIP-49 (i.e., the OHIP-14) consists of 14 items organized in the same seven sub-scales/domains, which address aspects of oral health that may compromise someone’s physical, psychological, and social well-being [16]. Three different scoring methods have been reported in studies using the OHIP-14: a summary OHIP-14 score (the sum of the seven raw sub-scale scores on a scale from 0 to 4, where a high score signifies worse OHRQoL); a weighted and standardized summary score (where weights are attributed to every question within the domain); and the total number of problems reported (i.e., occasionally, often, or very often, with a possible range of 0–14 problems) [16,23].

### 2.3. Dietary, Laboratory, and Clinical Assessment

Diet and eating habits were assessed with a validated food frequency questionnaire (FFQ) used in previous studies [19,24]. FFQ refers only to the frequency of intake and does not consider differences in portion sizes. The questionnaire investigates dietary habits over the previous year and inquiries about the consumption of 85 food items, which are further summarized in 28 food groups. The self-administered questionnaire was checked for completeness during an interview conducted by a physician at the study center. The questionnaire also includes questions about lifestyle aspects, such as educational level, physical activity, and smoking habits. Additionally, at the interview, anthropometric data on waist circumference (cm), weight (kg), and height (cm) were obtained. Weight and height were measured with the mechanical scale SECA 700 and stadiometer SECA 220 (Seca GmBH and Co., Hamburg, Germany), and the BMI was then derived and calculated as the ratio of weight (kg) to height squared (m^2^). The waist circumference was assessed with respect to the National Cholesterol Education Program: Adult Treatment Panel III (NCEP: ATP III) criteria. The WHO 2000 classification [25] was used to classify BMI into underweight (≤18.4 kg/m^2^), normal (18.5–24.9 kg/m^2^), overweight (25.0–29.9 kg/m^2^), and obese (≥30.0 kg/m^2^). In the present study, BMI was further classified into two groups, namely ideal and unfavorable, as follows: BMI values lower or equal 18.4 kg/m^2^ and over 30 kg/m^2^ were classified as unfavorable, while values between 18.5 and 30 kg/m^2^ were classified as ideal. For each individual, a blood sample was collected in the morning after overnight fasting to measure the levels of fasting blood glucose (FBG), total cholesterol, high-density lipoprotein cholesterol (HDL-C), and low-density lipoprotein cholesterol (LDL-C). The FBG level was measured using the glucose oxidase method (Sclavus), whereas the concentrations of plasma lipids (triglycerides, total cholesterol, and HDL-C) were quantified with an automated colorimetric device (Hitachi, Boehringer Mannheim). The LDL-C levels were measured using the Friedewald equation. Blood cell count was determined by a Coulter Hematology analyzer (Beckman–Coulter, Brea, CA, USA). The clinical assessment included extemporaneous ambulatory systolic blood pressure (SBP) and diastolic blood pressure (DBP), determined with the patient in a sitting position after at least a 10 min rest and taken at least 3 different times, using an automatic blood pressure monitor (Omron Healthcare). Diabetes mellitus and hypertension were diagnosed based on the following international diagnostic standards: FBG level higher than 125 g/dL (to convert to millimoles per liter, multiply by 0.0555) and SBP/DBP greater than or equal to 130/80 mm Hg.

### 2.4. Statistical Analysis

Continuous variables were expressed as mean ± standard deviation (SD), median (min to max) and categorical variables as the proportion (%). The whole sample was subdivided into two groups according to BMI categories (ideal BMI and unfavorable BMI) to describe important differences. The distribution of all variables was tested using the Shapiro distribution test. A *p* value equal or less 0.05 was chosen to define statistical significance. Logistic regression models were used to estimate the association effect between the unitary increases in the OHIP-14 total score as independent variables and ideal BMI (yes/no) as an outcome. To assess the confounding effect of a number of covariates, we built two hierarchical logistic regression models: the first unadjusted and the second model adjusted for all the covariates. The covariates were selected as confounders on the basis of associations with both the variable of interest (OHRQoL) and the outcome (BMI): age [17], sex [26,27], education [27,28], hypertension [29,30], carbohydrate consumption [6,31], and alcohol consumption [6,27]. Furthermore, in order to rank the domains/sub-scales of OHIP-14 that were most predictive for the ideal BMI condition, a random forest regression model was built on ideal BMI conditions (yes/no) as the output. The predictors considered were obtained from the OHIP-14 domains/sub-scales as follows: Domain 1(Functional limitation) (Question 1: Difficult to pronounce words plus Question 2: Worsened taste), Domain 2 (Physical pain) (Question 3: Pain plus Question 4: Uncomfortable to eat), Domain 3 (Psychological discomfort) (Question 5: Concern for the mouth plus: Question 6: Self-consciousness due to oral problems), Domain 4 (Physical disability) (Question 7: Diet unsatisfactory plus Question 8: Interrupted meals), Domain 5 (Psychological disability) (Question 9: Difficult to relax due to oral problems plus Question 10: Embarrassment due to oral problems), Domain 6 (Social disability) (Question 11: Irritability plus Question 12: Difficult to do jobs due to oral problems), Domain 7 (Handicap) (Question 13: Life less satisfying due to oral problems plus Question 14: Totally unable to function). We used the mean decrease in the Gini coefficient as a measure of how each OHIP-14 domain/sub-scale contributed to the homogeneity of the nodes and leaves in the resulting random forest regression model. In particular, Gini importance measures the average gain of purity by splits of a given variable. The higher the value of the mean decrease in the Gini score, the higher the importance of the OHIP-14 domains/subscale in the model. Statistical analysis was performed with RStudio software, Version 1.4.1106, using additional packages: Tidyverse, rstatistix, Epi, kableExtra, gmodels, randomForest, ggplot2.

## 3. Results

### 3.1. Descriptive Analysis

Table 1 summarizes the baseline sociodemographic, laboratory, and clinical variables of the whole sample (N = 216) according to the BMI status (ideal or unfavorable), including the nutritional assessment and the OHIP-14 assessment, which we have schematically and graphically shown in Figure 1. The ideal BMI group included 152 subjects (mean age = 71.95 ± 5.39 years) and was slightly dominated by females (47.4% males vs. 52.6% females). The unfavorable BMI group consisted of 64 subjects (mean age = 70.12 ± 4.05 years) with a higher presence of males (56.2%) than females (43.8%). Age was higher in the ideal BMI group (71.95 ± 5.39 years) than in the unfavorable BMI group (70.12 ± 4.05 years, *p* = 0.03), while DBP (*p* < 0.01), SBP (*p* < 0.01), FBG (*p* < 0.01), HbA1c (*p* < 0.01), and triglycerides (*p* = 0.05) were higher in the unfavorable BMI group compared to the ideal BMI group (Table 1). For the OHIP-14, the total score (*p* = 0.03), Question 4 (uncomfortable to eat, domain: physical pain) (*p* = 0.01), and Question 11 (irritability, domain: social disability) (*p* = 0.01) were more represented in the unfavorable BMI group if compared with the ideal BMI group (Table 1). Figure 2 shows the jitter box plot of the OHIP-14 total score across BMI status (ideal/unfavorable).

### 3.2. Logistic Regression Analyses

In Table 2, we showed two hierarchical logistic regression models used to estimate the association effect between the unitary increases in the OHIP-14 total score as independent variables and ideal BMI (yes/no) as an outcome. In the unadjusted model, an increase in the OHIP-14 total score increased the risk of an unfavorable BMI (odds ratio (OR): 1.08, 95% confidence interval (CI): 1.01–1.15, *p* = 0.03). In the model adjusted for age, sex, education, hypertension, carbohydrate consumption, and alcohol consumption, this finding was confirmed with an increase in the OHIP-14 total score that increased the risk to have an unfavorable BMI (OR: 1.10, 95% CI: 1.01–1.22, *p* = 0.04), and higher age linked to a decreased risk to have an unfavorable BMI (OR: 0.89, 95% CI: 0.82–0.97, *p* = 0.04).

### 3.3. Random Forest Regression Model

To rank the oral health domains/sub-scales of OHIP-14 that were the most predictive for the ideal BMI, we built a random forest regression model on the ideal BMI condition as the output. In the present study, the most important predictive domains/sub-scales of OHIP-14 in the mean decrease in the Gini coefficient for unfavorable BMI were, in order of decreasing importance, Domain 2 (Physical pain), Domain 1 (Functional limitation), Domain 3 (Psychological discomfort), the Domain 4 (Physical disability), the Domain 6 (Social disability), Domain 5 (Psychological disability), and finally, Domain 7 (Handicap) (Table 3). Figure 3 shows a dot chart of variable importance measured using the mean decrease in the Gini coefficient by the random forest regression model of the domains/sub-scales of the OHIP-14 with the ideal BMI status as output.

## 4. Discussion

In the present large population-based study on older adults from Southern Italy, negative OHRQoL, i.e., discomfort and disability attributed to oral conditions, increased the risk to have an unfavorable BMI in the hierarchical logistic regression models both unadjusted and also when adjusted for age, sex, education, hypertension, carbohydrate consumption, and alcohol consumption. Furthermore, higher age was linked to a decreased risk of an unfavorable BMI. The most important predictive domains/sub-scales of OHIP-14, measuring OHRQoL, for unfavorable BMI, were, in decreasing order of importance, physical pain, functional limitation, psychological discomfort, physical disability, social disability, psychological disability, and handicap.

The principal finding of the present study was that negative OHRQoL increased the risk to have an unfavorable BMI in a large population of older adults from Southern Italy, also after adjustment for a series of possible confounding factors. In recent years, in other population-based studies, the assessment of clinical oral indicators and OHRQoL has been widely used to evaluate whether oral problems may lead to nutritional dysfunction in older age [17,32,33]. In particular, Makhija and colleagues found a parabolic effect between OHRQoL and BMI in community-dwelling older adults living in Alabama, USA, with the strongest associations occurring in the underweight and obese categories [32]. These findings were similar to those of the present study in which BMI values < 18.4 kg/m^2^ (underweight) and >30 kg/m^2^ (obese) were classified as unfavorable, and higher OHRQoL increased the risk to have an unfavorable BMI. However, in the present study, we did not distinguish between older individuals who were underweight and overweight/obese. Moreover, Rosli and colleagues, using the GOHAI for measuring OHRQoL, found that subjects with a poor perception of their oral health were more likely to have an unfavorable BMI [17], with a classificatory system similar to that of the present study (older individuals underweight, overweight, and obese categorized as having an unfavorable BMI). Finally, Khongsirisombat and colleagues, in a hospital-based study on Thai older individuals, showed that those with obesity had an almost three times higher tendency to have a negative OHRQoL compared with the non-obese individuals [33]. Moreover, after adjusting for all related factors, the chances of predicting a prevalence of participants who scored four on at least one item on the OHIP-14 score based on obesity and oral dryness scores were 4.42 (95% CI:1.57–12.47) and 1.11 (95% CI:1.02–1.20), respectively. For every point of BMI or 1 cm increase in waist circumference, the chance of unfavorable OHRQoL also increased by a factor of 1.23 or 1.06, respectively, without the influence of xerostomia [33]. However, in some other cross-sectional studies, there was no association between OHRQoL measurements and nutritional status among older subjects [34,35]. These negative findings may be partly explained by various factors affecting food choices and intakes among older subjects, like general health, socioeconomic components, and taste and control over food preparation [36], with also a lack of knowledge on the nutritive value of foods consumed among older individuals, putting them still at risk of malnutrition regardless of their oral perceptions. Furthermore, the sample sizes in these previous negative studies were very small [34,35], without a comprehensive adjustment for possible confounders, and these factors could be a source of discrepancy with the present study.

Different studies showed an association between the impacts of OHRQoL and nutritional factors [37,38]. Older subjects with poor OHRQoL scores were shown to be at risk of nutritional deficiencies investigated with the Mini-Nutritional Assessment (MNA) [37,38], and not to have an actual unfavorable BMI. On the contrary, older adults with a better perception of oral health were among those at least at risk of malnutrition (lower MNA score) [39]. Both oral health and nutritional status are strongly related to healthy behaviors, and therefore these findings may also suggest that those who have poorer oral health may be less likely to be conscious about their diet. Furthermore, in the present study, higher age was linked to a decreased risk of an unfavorable BMI. This finding was consistent with the results from a Malaysian population-based study by Makhija and colleagues in which the number of older adults with an unfavorable BMI significantly decreased with advancing age [17]. This pattern may be explained by the fact that older individuals with an unfavorable BMI (i.e., obesity) may die earlier because of chronic diseases related to their condition like metabolic or cardiovascular diseases, thus leaving the non-obese individuals with a higher survival rate in the older age group.

In the present study, the OHRQoL domains more linked to physical manifestations (physical pain, functional limitations, and physical disability, with their items focusing on worsened taste, interrupted meals, and discomfort to eat) appeared to have a greater impact on nutritional factors associated with an unfavorable BMI compared to psychosocial manifestations of the OHRQoL (social disability and psychological disability, with their items focusing on the difficulty to relax or to do jobs due to oral problems). Moreover, the item “uncomfortable to eat” (domain: physical pain) and the item “irritability” (domain: social disability) were more represented in the unfavorable BMI group if compared with the ideal BMI group. In a recent hospital-based study, Khongsirisombat and colleagues, using OHIP-14 to evaluate OHRQoL, found that the average score was the highest for the physical pain domain in all studied groups [33]. Moreover, in the same study, all groups had the highest frequency of responses for the two items (item 3, pain, and item 4, uncomfortable to eat) of the physical pain domain of the OHIP-14 [33], suggesting that a possible underlying mechanism explaining why the higher rate of dental disease in patients with obesity/underweight might be a factor related to OHRQoL. The physical pain domain is determined by pain and discomfort when eating. The present findings are also consistent with a previous study in Norway showing that older individuals most frequently experienced problems with pain in the mouth and discomfort when eating [40]. Therefore, oral pain may lead to a negative OHRQoL in older adults who may experience difficulty chewing and swallowing due to a dry mouth, missing teeth, and dental and periodontal problems resulting in discomfort while eating and drinking [41,42].

The strengths of the present study were the population-based setting and the large number of older subjects included, notwithstanding a relatively small number of those investigated with the OHIP-14. However, given the cross-sectional nature of the study, we cannot make any inference on the direction of the association because of reverse causality; we can estimate the association only in terms of prevalence. Other studies showed no statistically significant differences in BMI between individuals with at least one tooth and persons with no teeth [32,43]. Therefore, dentate status was not included in the present analysis. Furthermore, the OHIP-14 items were self-reported and consequently, subjective. However, they provide important information on the perceptions of OHRQoL in older adults, and self-reported measures may be more meaningful than clinical measures in this context [32]. Finally, the present findings may not be generalizable; therefore, these results should be cautiously interpreted. Future studies in different counties should be performed with a larger sample size to collect additional data about OHRQoL and nutrition.

## 5. Conclusions

In the present large population-based study on older adults from Southern Italy, negative OHRQoL, i.e., discomfort and disability attributed to oral conditions, particularly related to the physical pain domain, increased the risk of being underweight or overweight and obesity also after adjustment for possible confounders. Furthermore, higher age was linked to a decreased risk of an unfavorable BMI. Therefore, the present study suggested that older adults with poor perception of oral health were more likely to have an unfavorable BMI. Moreover, a recent retrospective cohort study showed that poor oral health also with undernutrition could be used as an indicator to identify older nursing home residents at high risk for unplanned hospital visits [44]. The application of OHRQoL tools together with objective clinical oral indicators needs to be emphasized as it could be utilized as oral health predictors that might lead to impaired nutrition in older populations. These findings underlined the need for community interventions towards older subjects and their caregivers in order to improve general and oral health status. In fact, effective oral pain management and prevention of oral problems in older people may improve OHRQoL, and regular dental visits and treatment of oral disease can prevent weight gain or loss and poor self-perception of oral health [9].

## Figures and Tables

**Figure 1 jpm-13-01300-f001:**
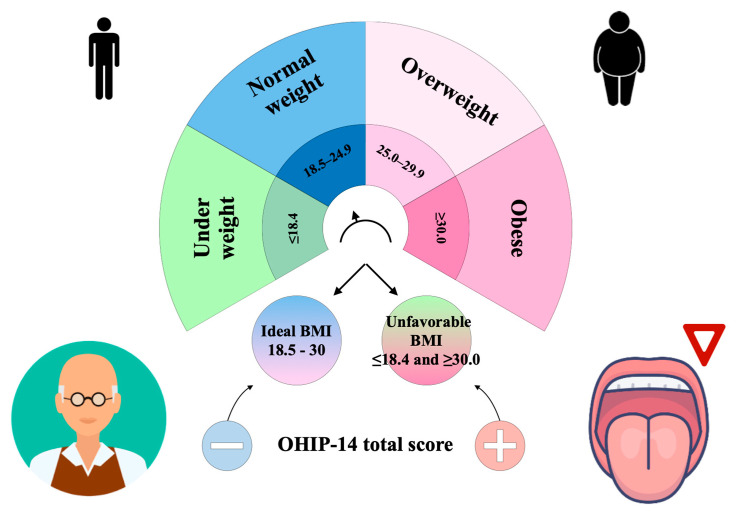
Schematic representation of the relationship between oral health-related quality of life measured with the Oral Health Impact Profile-14 (OHIP-14) questionnaire, and body mass index (BMI) in a population of older people.

**Figure 2 jpm-13-01300-f002:**
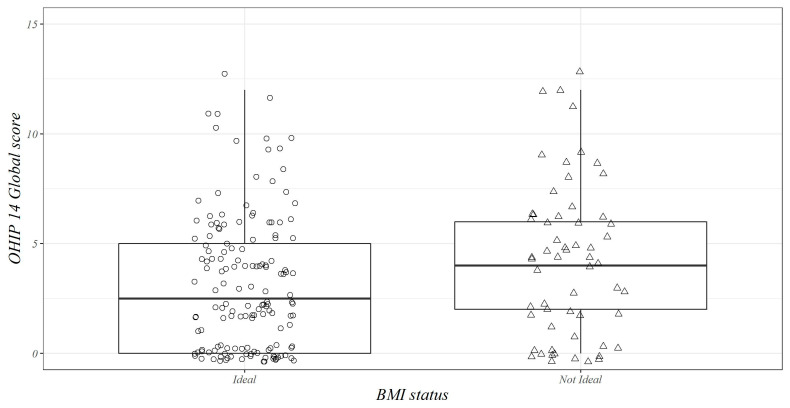
Jitter box plot of Oral Health Impact Profile-14 (OHIP-14) total score across body mass index (BMI) status (ideal/unfavorable). The Salus in Apulia Study (N = 216).

**Figure 3 jpm-13-01300-f003:**
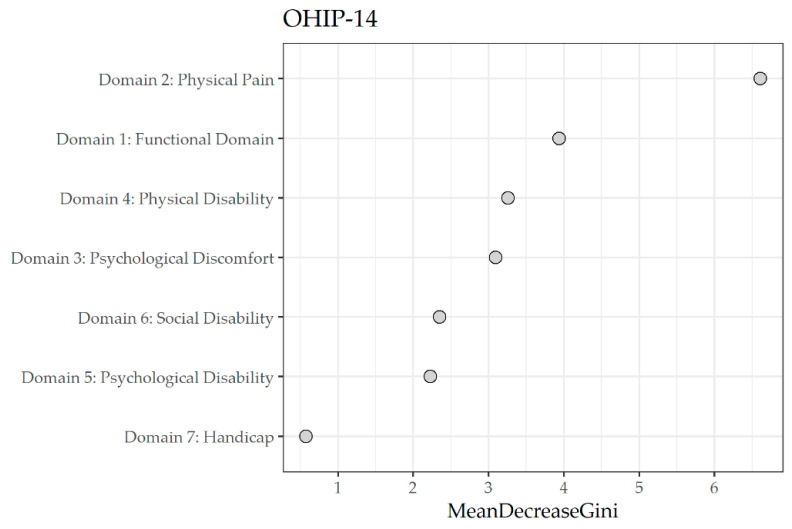
Dot chart of variable importance as measured by a random forest regression model of the domains/sub-scales of the Oral Health Impact Profile-14 (OHIP-14), with ideal body mass index (BMI) status as output. The Salus in Apulia Study (N = 216). Domain 1 (Functional limitation): Q1(Difficult to pronounce words) plus (Q2: Worsened taste). Domain 2 (Physical pain): Q3 (Pain) plus Q4 (Uncomfortable to eat). Domain 3 (Psychological discomfort): Q5 (Concern for the mouth) plus Q6 (Self-consciousness due to oral problems). Domain 4 (Physical disability): Q7 (Diet unsatisfactory) plus Q8 (Interrupted meals). Domain 5 (Psychological disability): Q9 (Difficult to relax due to oral problems) plus Q10 (Embarrassment due to oral problems). Domain 6 (Social disability): Q11 (Irritability) plus Q12 (Difficult to do jobs due to oral problems). Domain 7 (Handicap): Q13 (Life less satisfying due to oral problems) to Q14 (Totally unable to function).

**Table 1 jpm-13-01300-t001:** Sociodemographic, laboratory, and clinical variables of the whole sample according to the body mass index (BMI) status (ideal or unfavorable). The Salus in Apulia Study (N = 216).

	Ideal BMI		Unfavorable BMI		
	*Mean ± SD*	*Median* *(min to max)*	*Mean ± SD*	*Median* *(min to max)*	*p* ^§^
	Sociodemographic Assessment				
Proportion (%)	152 (70.40)		64 (29.60)		
Age (years)	71.95 ± 5.39	70 (65 to 87)	70.12 ± 4.05	70 (65 to 82)	**0.0**
Sex					
Male	72 (47.40)		36 (56.20)		0.23 *x*^2^
Female	80 (52.60)		28 (43.80)		
BMI (Kg/m^2^)	25.96 ± 2.84	26.42 (18.96 to 29.9)	33.7 ± 4.23	32.77 (17.38 to 47.69)	**<0.01**
Education (years)	7.39 ± 3.47	7 (0 to 18)	6.84 ± 3.38	5 (2 to 16)	0.21
	Nutritional Assessment				
Lipid consumption (g/die)	86.80 ± 52.00	78.60 (28.40 to 531)	85.40 ± 65.10	74.00 (33.20 to 501)	0.32
Carbohydrate consumption (g/week)	471.00 ± 227.00	449 (21.90 to 1382)	467.0 ± 342	402 (52.70 to 2344)	0.33
Protein consumption (g/die)	62.10 ± 24.10	59.50 (4.63 to 138)	86.10 ± 118.00	70.50 (33.20 to 887)	0.11
Alcohol consumption (g/die)	13.70 ± 18.70	10.40 (0 to 105)	16.30 ± 23.10	10.40 (0 to 81.10)	0.91
	Metabolic Biomarkers				
DBP (mmHg)	77.60 ± 6.54	80 (60 to 90)	81.40 ± 7.42	80 (60 to 100)	**<0.01**
SBP (mmHg)	130.00 ± 12.4	130.00 (100 to 160)	136.00 ± 15.4	140 (100 to 170)	**<0.01**
FBG (mg/dL)	98.41 ± 15.46	96 (70 to 166)	114.77 ± 33.96	105.5 (73 to 260)	**<0.01**
HbA1c (mmol/mol)	37.95 ± 7.38	37 (23 to 79)	42.62 ± 12.33	40.5 (28 to 101)	**<0.01**
Total cholesterol (mg/dL)	183.33 ± 36.59	180.5 (96 to 287)	182.22 ± 35.26	185 (76 to 248)	0.83 *
HDL cholesterol (mg/dL)	49.34 ± 13.16	46.5 (28 to 91)	45.61 ± 10.71	45 (27 to 74)	0.10
LDL cholesterol (mg/dL)	113.01 ± 30.45	111 (36 to 217)	113.7 ± 26.81	113.5 (55 to 182)	0.87 *
Triglycerides (mg/dL)	99.03 ± 44.7	92 (28 to 344)	113.78 ± 50.55	108 (39 to 261)	**0.05**
Hemoglobin (g/dL)	13.88 ± 1.3	13.9 (10.3 to 16.9)	13.94 ± 1.25	13.7 (11.4 to 16.5)	0.87
RBC (10^6^ cells/mm^3^)	5.01 ± 2.96	4.75 (3.58 to 40.8)	4.79 ± 0.47	4.76 (3.95 to 6.01)	0.88
WBC (10^3^ cells/mm^3^)	6.1 ± 1.75	5.9 (2.6 to 10.7)	5.97 ± 1.5	5.9 (3.06 to 9.4)	0.77
Platelets (10^3^ cells/mm^3^)	224.74 ± 54.74	219.5 (114 to 459)	231.78 ± 60.75	234.5 (110 to 452)	0.24
	OHIP-14 questionnaire				
Q1 Difficult to pronounce words	0.10 ± 0.44	0 (0 to 3)	0.13 ± 0.50	0 (0 to 2)	0.72
Q2 Worsened taste	0.24 ± 0.70	0 (0 to 4)	0.31 ± 0.82	0 (0 to 3)	0.75
Q3 Pain	1.14 ± 1.12	2 (0 to 4)	1.34 ± 1.13	2 (0 to 3)	0.20
Q4 Uncomfortable to eat	1.02 ± 1.30	0 (0 to 4)	1.50 ± 1.35	2 (0 to 4)	**0.01**
Q5 Concern for the mouth	0.16 ± 0.60	0 (0 to 3)	0.31 ± 0.88	0 (0 to 4)	0.18
Q6 Self-consciousness due to oral problems	0.26 ± 0.69	0 (0 to 3)	0.31 ± 0.84	0 (0 to 4)	0.96
Q7 Diet unsatisfactory	0.12 ± 0.56	0 (0 to 4)	0.21 ± 0.68	0 (0 to 3)	0.24
Q8 Interrupted meals	0.11 ± 0.43	0 (0 to 2)	0.21 ± 0.68	0 (0 to 3)	0.30
Q9 Difficult to relax due to oral problems	0.17 ± 0.55	0 (0 to 3)	0.11 ± 0.48	0 (0 to 3)	0.42
Q10 Embarrassment due to oral problems	0.11 ± 0.52	0 (0 to 3)	0.20 ± 0.70	0 (0 to 4)	0.31
Q11 Irritability	0.01 ± 0.16	0 (0 to 2)	0.13 ± 0.53	0 (0 to 3)	**0.01**
Q12 Difficult to do jobs due to oral problems	0.01 ± 0.16	0 (0 to 2)	0.08 ± 0.45	0 (0 to 3)	0.14
Q13 Life less satisfying due to oral problems	0.01 ± 0.08	0 (0 to 1)	0.06 ± 0.08	0 (0 to 3)	0.14
Q14 Totally unable to function	0.00 ± 0.00	0 (0 to 0)	0.03 ± 0.25	0 (0 to 2)	0.11
OHIP-14 Total score	3.46 ± 3.75	2.5 (0 to 25)	5 ± 5.67	4 (0 to 37)	**0.03**

All data are shown as mean ± standard deviation (SD), median (min to max) for continuous variables and n (%) for proportions, ^§^ Mann–Whitney U test where not otherwise specified, * independent samples *t* test, χ^2^ chi squared test, DBP: diastolic blood pressure; SBP: systolic blood pressure; FBG: fasting blood glucose; HbA1c: glycated hemoglobin; HDL: high-density lipoprotein; LDL: low-density lipoprotein; RBC: red blood cells, WBC: white blood cells, OHIP-14: Oral Health Impact Profile-14.

**Table 2 jpm-13-01300-t002:** Hierarchical logistic regression models on the body mass index (BMI) status (ideal/unfavorable) as dependent variables and the Oral Health Impact Profile-14 (OHIP-14) total score as regressor. The Salus in Apulia Study (N = 216).

	OR	95% CI	*p* Value
	Model 1		
OHIP-14 total score	1.08	1.01 to 1.15	**0.03**
	Model 2		
OHIP-14 total score	1.10	1.01 to 1.22	**0.04**
Age (years)	0.89	0.82 to 0.97	**<0.01**
Sex (Female)	0.48	0.21 to 1.11	0.08
Education (years)	0.91	0.81 to 1.02	0.11
Carbohydrates consumption (g/week)	1.10	0.95 to 1.10	0.39
Alcohol consumption (g/day)	1.00	0.98 to 1.02	0.80

OR: odds ratio; CI: confidence interval.

**Table 3 jpm-13-01300-t003:** Random forest regression model of importance of the domains/sub-scales of the Oral Health Impact Profile-14 (OHIP-14) with ideal body mass index (BMI) status as output. The Salus in Apulia Study (N = 216).

Domain	Mean Decrease in Gini
2 Physical pain	6.63
1 Functional limitation	3.54
3 Psychological discomfort	3.15
4 Physical disability	3.04
6 Social disability	2.54
5 Psychological disability	2.28
7 Handicap	0.5

Domain 1 (Functional limitation): Q1 (Difficult to pronounce words) plus (Q2: Worsened taste). Domain 2 (Physical pain): Q3 (Pain) plus Q4 (Uncomfortable to eat). Domain 3 (Psychological discomfort): Q5 (Concern for the mouth) plus Q6 (Self-consciousness due to oral problems). Domain 4 (Physical disability): Q7 (Diet unsatisfactory) plus Q8 (Interrupted meals). Domain 5 (Psychological disability): Q9 (Difficult to relax due to oral problems) plus Q10 (Embarrassment due to oral problems). Domain 6 (Social disability): Q11 (Irritability) plus Q12 (Difficult to do jobs due to oral problems). Domain 7 (Handicap): Q13 (Life less satisfying due to oral problems) to Q14 (Totally unable to function).

## Data Availability

The data presented in this study are available upon request to the corresponding Authors Francesco Panza (email: f_panza@hotmail.com) and Vittorio Dibello (email: vittoriodibello1@gmail.com).

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
