# Peer review of "The Relationship between Oral Health-Related Quality of Life and Body Mass Index in an Older Population from Southern Italy: The Salus in Apulia Study"

_jpm, 2023, doi:10.3390/jpm13091300_

Round 1
Reviewer 1 Report
The paper is well developed and the objectives are well defined for this kind of research. However, I recommend some minor corrections in order to improve the manuscript before publication.
In the abstract, the authors have "The oral health-related quality of life (OHRQoL) assessment evaluated the impact of individual’s oral health on the patient's physical and psychosocial status". Assessment and evaluation have the same meaning. The authors should refer "The oral health-related quality of life (OHRQoL) scale evaluated the impact of individual’s oral health on the patient's physical and psychosocial status". Also in the abstract: "In a random forest regression model, the most important predictive domains/subscales of OHIP-14 in mean decrease Gini for unfavorable BMI were, in order of decreasing...". I believe there should me a mistake in the sentence that should be corrected and that the authors should be referring to the Gini coefficient.
In the Results section, I recommend a better and more complete interpretation of the results presented in the tables and graphs.
In the Discussion section, the authors should refer some of the main authors of the scientific articles referenced along the manuscript.
The authors refer ar the end of the Discussion section that "Finally, the present findings may not be generalizable; therefore, these results should be cautiously interpreted." This should be more developed and explained in what way the results can be improved.
In the conclusions, the authors should refer to the need for community interventions towards the elderly and their caregivers in order to improve general and oral health status.
English should be improved, mainly changing to more technical terms. For example, change evaluation to assessment.
Author Response
Manuscript reference number: jpm-2575086
Title: The Relationship between Oral Health-Related Quality of Life and Body Mass Index in an Older Population from Southern Italy: The Salus in Apulia Study
Reviewer #1:
The paper is well developed and the objectives are well defined for this kind of research. However, I recommend some minor corrections in order to improve the manuscript before publication.
1. In the abstract, the authors have "The oral health-related quality of life (OHRQoL) assessment evaluated the impact of individual’s oral health on the patient's physical and psychosocial status". Assessment and evaluation have the same meaning. The authors should refer "The oral health-related quality of life (OHRQoL) scale evaluated the impact of individual’s oral health on the patient's physical and psychosocial status". Also in the abstract: "In a random forest regression model, the most important predictive domains/subscales of OHIP-14 in mean decrease Gini for unfavorable BMI were, in order of decreasing...". I believe there should me a mistake in the sentence that should be corrected and that the authors should be referring to the Gini coefficient.
1. We thank the Reviewer for this comment. However, OHRQoL is not a scale but a kind of assessment. We modified as follows the sentence: “The assessment of oral health-related quality of life (OHRQoL) evaluated the impact of individual’s oral health on the patient's physical and psychosocial status”. We modified as follows also the other sentence: “In a random forest regression model, the most important predictive domains/sub-scales of OHIP-14 in mean decrease in Gini coefficient for..”.
2. In the Results section, I recommend a better and more complete interpretation of the results presented in the tables and graphs.
2. We described the principal and statistically significant findings of the descriptive analysis, the logistic regression analyses, and the random forest regression model. We modified for clarity some sentences in this revised version of the manuscript.
3. In the Discussion section, the authors should refer some of the main authors of the scientific articles referenced along the manuscript.
3.Point fixed.
4. The authors refer ar the end of the Discussion section that "Finally, the present findings may not be generalizable; therefore, these results should be cautiously interpreted." This should be more developed and explained in what way the results can be improved.
4. We thank the Reviewer also for this comment. We included a statement on this (please, see page 13).
5.In the conclusions, the authors should refer to the need for community interventions towards the elderly and their caregivers in order to improve general and oral health status.
5. We included this thoughtful consideration at the end of the Conclusions section (plaese, see page 14).
5. Comments on the Quality of English Language
English should be improved, mainly changing to more technical terms. For example, change evaluation to assessment.
5. We revised the whole manuscript, according the suggestions of this Reviewer.
Reviewer 2 Report
The research is relevant and well done, but there are a few questions that clarify.
1. How was the sample size determined? Only according to the criteria that you derived or was a preliminary calculation of the sample size determined?
2. Did you receive permission from the ethical committee for the study?
3. Please improve the quality of figure 2.
4- Please write the figure caption in the below section of figures.
5. According to the list of references, some new and more relevant references can be added.
Author Response
Manuscript reference number: jpm-2575086
Title: The Relationship between Oral Health-Related Quality of Life and Body Mass Index in an Older Population from Southern Italy: The Salus in Apulia Study
Reviewer #2:
The research is relevant and well done, but there are a few questions that clarify.
1. How was the sample size determined? Only according to the criteria that you derived or was a preliminary calculation of the sample size determined?
1. We thank very much for this thoughtful comment of the Reviewer. The sample size was determined according the criteria described in the manuscript (please, pages 3 and 4).
2. Did you receive permission from the ethical committee for the study?
2. We thank this Reviewer for the comment. As indicated in the section "Institutional Review Board Statement" at the end of the manuscript (page 14), this study was approved by the Institutional Review Board of the National Institute of Gastroenterology “S. De Bellis”, Castellana Grotte, Bari, Italy. In this revised version of the manuscript, we added this statement also in the Materials and Methods section (please, see page 3).
3. Please improve the quality of figure 2.
3. We thank again this Reviewer for the useful suggestion. In this revised version of the manuscript, the quality of figure 2 has been improved.
4. Please write the figure caption in the below section of figures.
4. We modified the figures as requested by the Reviewer by adding captioning below them.
5. According to the list of references, some new and more relevant references can be added.
5. In this revised version of the manuscript, we included a new and relevant reverence as suggested by this Reviewer (Reference 44).